# The Exon Junction Complex and intron removal prevent re-splicing of mRNA

**Brian Joseph**[1,2], **Eric C. Lai**[1,2]*

**1** Developmental Biology Program, Sloan Kettering Institute, New York, New York, United States of America,
**2** Louis V. Gerstner, Jr. Graduate School of Biomedical Sciences, Memorial Sloan Kettering Cancer Center, New York, New York, United States of America

* laie@mskcc.org

## Abstract

Accurate splice site selection is critical for fruitful gene expression. Recently, the mammalian EJC was shown to repress competing, cryptic, splice sites (SS). However, the evolutionary generality of this remains unclear. Here, we demonstrate the *Drosophila* EJC suppresses hundreds of functional cryptic SS, even though most bear weak splicing motifs and are seemingly incompetent. Mechanistically, the EJC directly conceals cryptic splicing elements by virtue of its position-specific recruitment, preventing aberrant SS definition. Unexpectedly, we discover the EJC inhibits scores of regenerated 5' and 3' recursive SS on segments that have already undergone splicing, and that loss of EJC regulation triggers faulty resplicing of mRNA. An important corollary is that certain intronless cDNA constructs yield unanticipated, truncated transcripts generated by resplicing. We conclude the EJC has conserved roles to defend transcriptome fidelity by (1) repressing illegitimate splice sites on pre-mRNAs, and (2) preventing inadvertent activation of such sites on spliced segments.

**Data Availability Statement:** All custom scripts used in this study are reported on the Lai lab GitHub page (https://github.com/Lai-Lab-Sloan-Kettering).

**Funding:** Work in E.C.L.'s group was supported by the National Institutes of Health (R01-NS083833

## Author summary

The Exon Junction Complex (EJC) is a conserved multiprotein complex that is deposited ~20–24 nucleotides upstream of exon-exon junctions during mRNA splicing. Although the EJC is well-conserved, many of its overt regulatory requirements differ between species. For example, the mammalian EJC is involved in mRNA surveillance and nonsense mediated decay (NMD), and also suppresses cryptic splicing. On the other hand, the *Drosophila* EJC does not mediate NMD, and it has multiple roles in promoting splicing of long introns and suboptimal splicing substrates. Here, we unify this by showing that the *Drosophila* EJC suppresses splicing at hundreds of illegitimate cryptic splice sites, which are presently unannotated in the well-studied *Drosophila* genome. As in mammals, this role takes advantage of the sequence-independent deposition of the EJC upstream of splice sites, and appears to represent an ancestral function. We expand this concept by showing the necessity of the EJC to prevent resplicing in exonic remnants that inherently regenerate splice sites following canonical splicing. Importantly, cDNA expression constructs evade EJC regulation, and we show that utilization of cDNAs can unintentionally trigger re-splicing into unanticipated products with internal deletions.

and R01-GM083300) and MSK Core Grant P30-
CA008748. The funders had no role in study
design, data collection and analysis, decision to
publish, or preparation of the manuscript.

**Competing interests:** The authors have declared
that no competing interests exist.

## Introduction

Canonical splice sites contain instructive information across the exon/intron boundary. Pioneering studies using genetics and biochemistry demonstrated that U1 snRNA establishes contacts with 5' SS, AG|GUAAGU (where | marks the exon/intron boundary) [1–4]. Similarly, the U2AF complex shows preference for a AG|GU motifs at 3' SS, which includes two nucleotides into the exon [5]. Moreover, exonic segments of splice sequences are also utilized during the catalytic stages of splicing, for example the juxtaposition of exon boundaries by U5 snRNA [6,7]. Thus, when processed, exon junctions contain remnants of splice sequences. However, the activity of these segments post-splicing remains little explored.

It has been observed that exon junction sequences can function as cryptic splice sites [8,9]. This has led to one view of intron birth, in which they insert into cryptic or protosplice sites; sequences that are typically inactive but contain the information content required to pair with spliceosomal building blocks, such as U1 snRNP or U2AF [2,5]. However, an alternate assessment is that intron removal may regenerate cryptic splice sites. This has been observed at cassette exons in the context of recursive splicing, but the recent discovery of suppressed, 5' recursive splice sites at constitutive exon junctions [10,11] reignites this discussion by suggesting that even seemingly constitutive exons may regenerate cryptic splice sites at exon junctions. Furthermore, these studies showed that recruitment of the Exon Junction Complex (EJC) silences the activity of cryptic splice sites [10,11].

The EJC is a multisubunit conglomerate that is deposited in a sequence-independent fashion ~24 nt upstream of exon-exon junctions [12,13]. Assembly of its three-member core complex begins during splicing, and the first step involves the position-specific deposition of the DEAD-box protein eIF4AIII onto RNA by the spliceosome factor CWC22. Next, a heterodimer of MAGOH/Mago Nashi and RBM8A/Y14/Tsunagi binds eIF4AIII, stabilizing the complex on RNA. The core EJC complex interacts with multiple peripheral complexes involved in diverse RNA metabolism pathways [14]. Accordingly, EJC dysfunction broadly affects development, disease and cancer [15].

Curiously, while the EJC is well-conserved, the literature indicates fundamental differences in its requirements between invertebrates and vertebrates [14]. The EJC was first linked to the process of nonsense mediated mRNA decay [16,17], a process that exploits deposition of the EJC by the spliceosome [18]. Translation removes EJCs from the open reading frame, but the presence of premature termination codons cause EJCs to remain within aberrant 3' UTRs, thereby triggering NMD. However, as introns do not inherently elicit NMD in *Drosophila*, its pathway does not appear to involve the EJC [19].

Physical associations between the EJC and the spliceosome [20] have also warranted attention towards splicing-related functions of the EJC. Here, as well, there is evidence for functional distinctions amongst species. In *Drosophila*, the EJC positively regulates splicing of long introns, such as *mapk* [21,22], and also activates suboptimal splice sites, such as within *piwi* [23,24]. By contrast, recent analysis of the mammalian EJC shows that many of its direct splicing targets are instead inhibited [10,11], indicating a role in cryptic splice site avoidance during pre-mRNA maturation. However, the generality and scope of such a mode of splicing control has not been addressed. Overall, the available data suggest divergent impacts of the EJC on splicing, either promoting (in *Drosophila*), or inhibiting (in mammals) this fundamental mRNA processing process.

Here, we analyze the effects of the EJC on splicing in flies in detail. Although *Drosophila melanogaster* has one of the best annotated metazoan transcriptomes [25–27], we unexpectedly detect many hundreds of novel splice junctions upon depletion of core EJC components in a single celltype. *De novo* splicing analysis demonstrates the fly EJC protects neighboring introns

from cryptic splice site activation, as in mammals This function is required under unusual circumstances including out-of-order splicing and appears to rely on occlusion of competing, weak splice sites. Next, we identify scores of splice defects that arise from cryptic splice sites at exon junction sequences. Two key sources of evidence implicate exon junction sequences as sources of cryptic splice sites. First, we validate that cryptic splice donors and acceptors are regenerated at exon junctions. Second, we elucidate that even poor matches to consensus splice motifs can act as functional splice sites at exon junctions. While these sites are suppressed on pre-mRNAs, we find that silencing is also required on mRNAs to prevent further resplicing. Our results suggest that exon junction sequences are a source of cryptic 5' and 3' SS, and provides the basis for an intrinsic requirement of the EJC to suppress accidental activation. Overall, our findings broaden a newly appreciated, ancestral function of the EJC, and emphasize that bypass of this regulatory process via cDNA constructs can have unexpected deleterious consequences.

## Results

### EJC depletion leads to activation of spurious junctions

Recently, Roignant and colleagues reported RNA-seq datasets from S2 cells depleted for core EJC factors *eIF4AIII*, *tsu* (Y14) and *mago* [28]. We re-examined these data for splicing defects, and paid particular attention to spurious splice site usage. We utilized MAJIQ to acquire currently unannotated junctions (3606 novel splice sites supported by ≥5 split reads in the aggregate data), of which 1677 were >2-fold upregulated in at least one EJC-KD condition. As the three core EJC factors are mutually required for stable EJC association at exon-exon junctions, we might expect these to reveal a set of common molecular defects. Indeed, there was both substantial and significant overlap in novel junctions amongst all three conditions (*p-value* < $1x10^{-8}$ for three-way overlap), and 876 junctions were elevated in two out of three EJC-KD datasets (**S1A Fig**). To introduce further stringency, we also filtered for >2-fold PSI change in 2/3 EJC depletions, yielding 573 spurious junctions from 386 genes (**S1B Fig** and **S1 Table**). These genes are diverse, with gene ontology (GO) analysis comprising diverse cellular processes including system development and signaling (**S2 Table**).

The most frequent spurious junctions involved activation of alternative 5' or 3' SS that mapped to exons of canonical transcripts–we refer to these sites as exonic 5' SS or 3' SS. The other major classes were novel alternative splicing and activation of SS that mapped to introns (intronic SS) (**Fig 1A**). These are expected to delete exonic sequence (alternative 5' or 3' SS) or insert intronic sequence (intronic SS), relative to canonical mRNA products. We depict *straw* as an example of aberrant splicing occuring at a constitutive exon-exon junction (**Fig 1B**). Here, depletion of *eIF4AIII*, *tsu* and *mago*, but not *lacZ* control, all induced high-frequency usage of a novel exonic, alternative 5' SS that joins to the constitutive 3' SS 3248 nt downstream. Importantly, this presumably defective transcript comprises the major isoform in all three core-EJC knockdowns, as it removes 91 nt of coding sequence and is thus out of frame. An additional example of a cryptic exonic 3' SS from the *mask* gene is shown in **Fig 1C**.

We used rt-PCR to validate *de novo* splice isoforms in EJC-depleted S2 cells, focusing on the dominant classes of cryptic exonic splice site usage (**Fig 1A**). For these cases, aberrant usage of cryptic exonic splice sites, either at the 5' or the 3' end, will yield internally truncated products that can be distinguished from longer wildtype products (**Fig 1D**). We selected transcripts with high activation of exonic 5' and 3' SS (PSI > 0.2), such as *straw, multiple ankyrin repeats single KH domain (mask), baboon* and *eukaryotic translation initiation factor 4G1 (eIF4G1)*, but also evaluated targets with moderate changes (0.01< PSI < 0.05) such as *Crk oncogene and unkempt*. As EJC stabilization during pre-mRNA processing requires *eIF4AIII*,

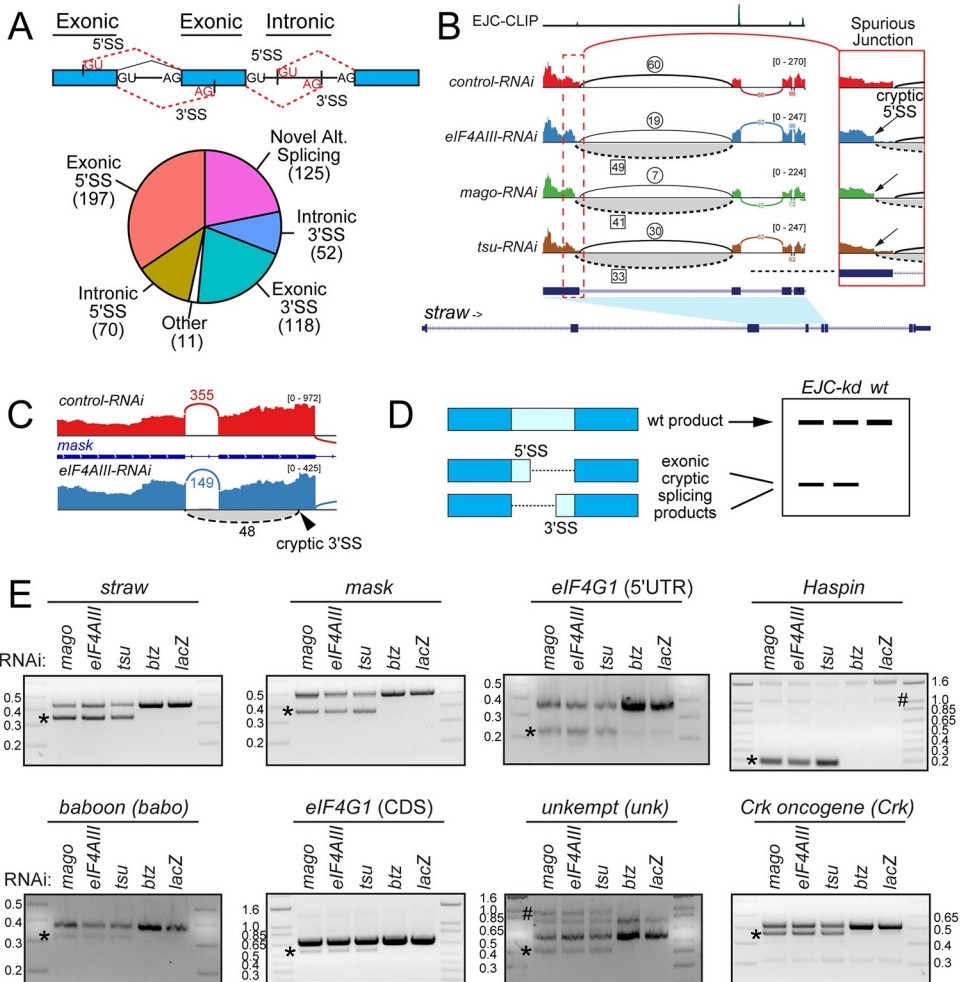

**Fig 1. Transcriptome-wide *de novo* alternative splicing upon depletion of functional Exon Junction Complex.** (A) Overview of upregulated *de novo* splice junctions in EJC-depleted cells. Top: schematic of cryptic 5' and 3' SS. In this toy gene model, canonical pre-mRNA exons and introns are depicted as blue boxes and black lines. The ends of these introns are marked by splice signatures (GU: donor and AG: acceptor, shown in black). Cryptic splice sites identified in the EJC LOF datasets can be found within sequences that are normally exonic or intronic. These sites and the putative *de novo* intron are shown as red text and dashed lines. Bottom: Pie chart indicating the distribution of different splice junction classes. (B) Sashimi plot depicting HISAT2-mapped sequencing coverage along a portion of *straw*, which has defective splicing under core-EJC LOF. The gene model depicts the location of the cryptic 5' SS relative to the annotated 5' SS. Junction spanning read counts mapping to the canonical junction are circled, whereas cryptic junction read counts are squared. Note that spliced reads mapping to the cryptic junction are found in *eIF4AIII*-, *mago*- and *tsu*-KD but not the control comparison. EJC-CLIP [29] shows recruitment of EJC to exon-exon junctions. Region containing the cryptic 5' SS has been zoomed on the right. (C) Sashimi plot depicting RNA-seq coverage at the mask gene, where depletion of the EJC (shown here, *eIF4AIII-RNAi*) results in utilization of a cryptic exonic 3' SS. (D) Schematic for rt-PCR validation of exonic cryptic splicing, which yields shorter, internally deleted products. (E) Validation of *de novo* splicing events in core-EJC depleted cells. EJC core components (*eIF4AIII*, *mago*, *tsu* and *btz*) were depleted from *Drosophila* S2 cells using dsRNA. After knockdown, eight targets identified in (A) were evaluated using rt-PCR and demonstrated splicing defects (asterisk). Importantly, only knockdown of core-EJC factors yielded cryptic bands, but not *btz* or control conditions. *unkempt (unk)* generates several products due to multiple alternative exons included within its rt-PCR amplicon (S1D Fig). # indicates products of unknown identity.

*tsu* and *mago*, but not *btz*, we utilized knockdown of *btz* and *lacZ* as controls (**S1C Fig**). We designed rt-PCR primers that mapped to each mRNA sequence with the cryptic splicing event nested within each amplicon (**S2 Fig**). For all eight genes tested, we observed splicing defects only under core-EJC (*eIF4AIII*, *tsu* and *mago*) knockdown conditions (**Fig 1C**). We note that

our rt-PCRs proved sensitive enough to detect products that appeared abundant as well as those that were lowly expressed. In the case of *unk*, we detected the known complement of annotated alternative splice isoforms, but also detected the expected spurious product under EJC LOF (**Figs 1C** and **S1D**). These data provide stringent validation of our annotation of spurious junctions, and highlight a previously unappreciated quality control function of the *Drosophila* EJC.

## The EJC suppresses cryptic exonic 3' SS during pre-mRNA processing

These alterations in transcript processing were reminiscent of how the human EJC, recruited to exon junctions, directly influences the splicing of neighboring introns [10,11]. Accordingly, we examined the mechanism of EJC-regulated splicing defects in *Drosophila*. We began by examining transcripts with spurious 3' SS that mapped to exons on wildtype transcripts. These represent a large proportion of *de novo* events observed in our analysis, and are predicted to cause broad loss of mRNA sequences. Cryptic 3' SS exhibit strong positional bias and cluster specifically around canonical exon junctions (**Fig 2A**). However, while cryptic 3' SS contain the invariant 3' AG dinucleotide (**S3A Fig**), quantitative assessment of SS strength indicated broad variation (**Fig 2A**). In fact, most activated 3' SS in this category are extremely weak and would not normally be considered functionally competent, especially when considering their sheer frequency in the transcriptome at large. We also considered a scenario in which weak cryptic 3' SS might be supported by strong branch point sequences (BP). To evaluate this, we first bioinformatically derived a branch point motif by analyzing intronic sequences upstream of canonical *Drosophila* 3' SS (**S3B Fig**). We then assessed the presence and quality of potential BP motifs 15-45 nt upstream of spurious 3' SS, and found that the distribution of BP motifs was similar between weak and strong cryptic splice sites (**S3C Fig**). Notably, a substantial fraction of splice sites derepressed in EJC-depleted cells bore weak 3' SS and lacked overt BPs. Thus, it was important to manipulate these RNA substrates to understand their splicing capacities more directly.

We selected *CG7408* as a paradigm: it reproducibly exhibited defective splice isoforms in all core-EJC knockdowns (**Fig 2B**), but its putative 3' SS is extremely weak (NNSPLICE score of 0.29, **Fig 2A**) and poorly conserved (**S3D Fig**). We used rt-PCR to validate the expected transcript defects in EJC-depleted cells (**Fig 2C**). The wildtype splicing pattern of this locus (intron 1) involves three alternative 3' SS (A3'SS), all of which are activated in S2 cells as evidenced by RNA sequencing as well as rt-PCR (**Figs 2B, 2C and S3D**). The dominant isoform yields the longest product and utilizes an annotated 3' SS that is stereotypically strong in comparison to the cryptic 3' SS (NNSPLICE score of 0.91). Activation of the spurious 3' SS, which lies downstream of all three annotated A3'SS, yields a product lacking 183 nt relative to the dominant wildtype transcript. We constructed a minigene bearing exons 1–4 of wildtype *CG7408*, allowing us the opportunity to explore aberrant processing (**Fig 2D**, genomic). When transfected into S2 cells, this reporter produced transcripts that indicated use of annotated A3'SS (**Fig 2E**, genomic and **Fig 2C**, *lacZ*). Importantly, a reporter lacking all introns, i.e., mimicking an mRNA expression construct, yielded a single wildtype product (**Fig 2D and 2E**, mRNA). Thus, intronless *CG7408* transcripts that cannot recruit EJC, also do not undergo further processing.

Next, we explored our minigene reporter by testing for potentially distinct consequences of EJC recruitment to individual *CG7408* exon junctions, by deleting each intron in turn (**Fig 2D**—Δi1, Δi2 and Δi3). These manipulations should only abolish EJC recruitment at individual exon junctions that do not require splicing due to intron deletion. Δi1 only produced wildtype transcript and Δi3 produced wildtype isoforms at the same proportions as the genomic construct (**Fig 2E**). By contrast, deletion of intron 2 yielded aberrant transcripts through

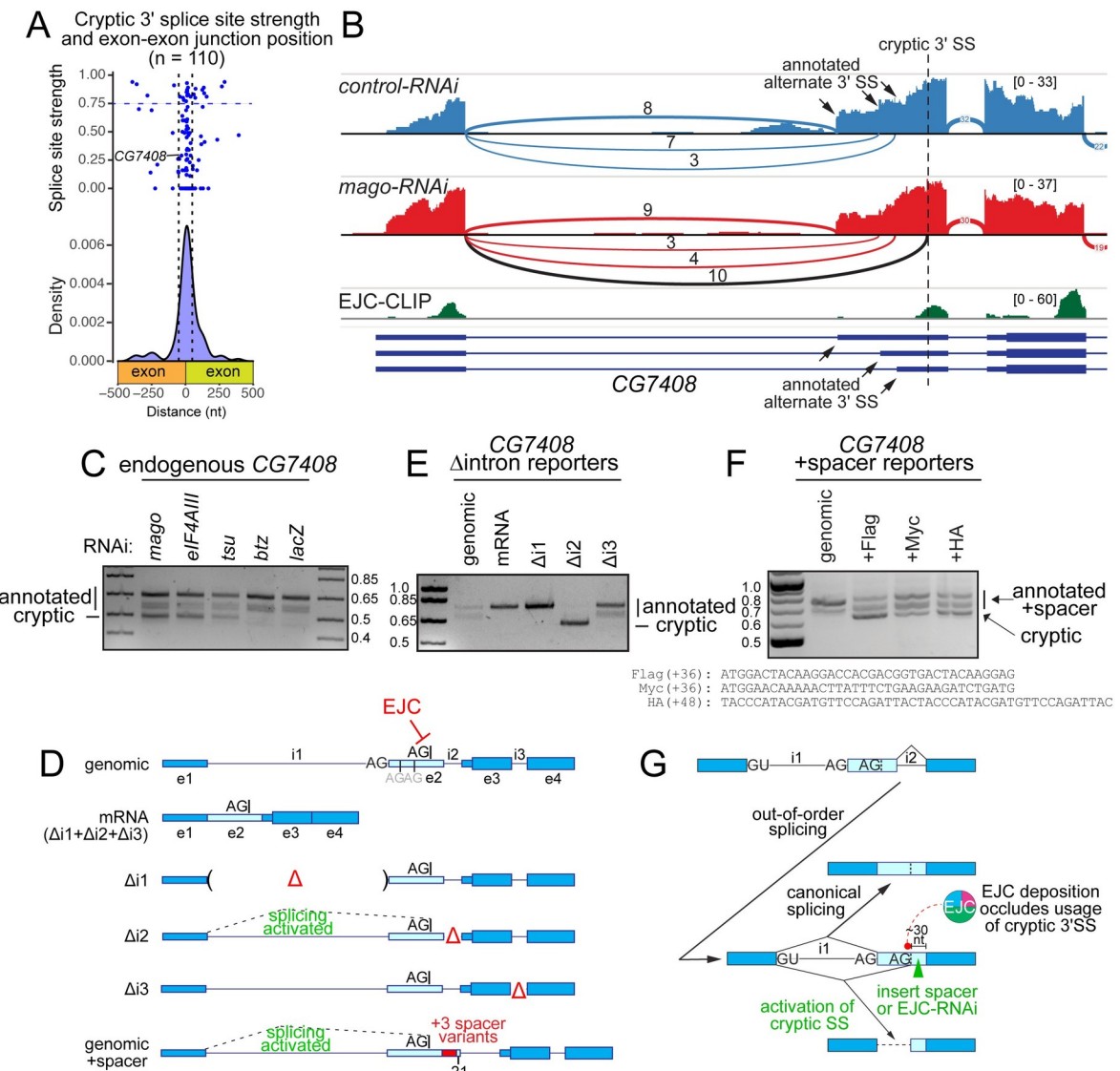

**Fig 2. EJC-depletion leads to activation of cryptic 3' splice sites.** (A) Depiction of 3' SS position of spurious junctions relative to exon-exon boundaries as density and dot plot. The dot plot indicates splice site scores as calculated via NNSPLICE. Horizontal dashed line depicts threshold for strong 3' SS, and vertical dashed lines specify 50 nt flanking exon-exon junctions. (B) Sashimi plot depicting HISAT2-mapped sequencing coverage along a portion of *CG7408*, which has a cryptic 3' SS that is activated under core-EJC LOF. Junction spanning read counts mapping to the canonical junction are circled, whereas cryptic junction read counts are squared. Note that spliced reads mapping to the cryptic junction are found in *eIF4AIII*, *mago* and *tsu* KD but not the control comparison. EJC-CLIP [29] shows recruitment of EJC to exon-exon junctions. (C) Validation of *CG7408* cryptic 3' SS activation in core-EJC, but not *btz* or *lacZ* KD conditions. (D) Schematic of *CG7408* splicing reporters. Exons 1–4 (introns included) were cloned and subjected to further manipulation. Locations of deleted introns (Δ), as well as a construct lacking all introns (mRNA) are included. For reference, the position of the cryptic 3' SS is marked on exon 2. genomic+spacer represents modified versions of the genomic splicing reporter with insertions of 36–48 nt spacer sequences on exon 2. (E) rt-PCR of reporter (D) constructs ectopically expressed in S2 cells demonstrates that intron 2 is required for accurate processing of the minigene. Canonical and cryptic products are indicated. (F) Cryptic splicing is detected with the inclusion of 36–48 nt spacer sequences. (G) Schematic of out-of-order splicing and positional requirement of the core-EJC for accurate 3' SS definition.

activation of the same weak spurious 3' SS identified in core-EJC knockdown datasets (**Fig 2E**, Δi2). These tests emphasize a functional requirement of intron 2 for correct processing of *CG7408* and demonstrate that even poor matches to consensus splice sites (i.e., the *CG7408* cryptic 3' SS) can be potently activated in the absence of the EJC.

How is the EJC involved in suppression of cryptic 3' SS near canonical exon junctions? In human cells, the EJC can directly mask cryptic 3' SS. Based on the close clustering of these sites around the position of EJC recruitment (**Fig 2A**), we reasoned that *Drosophila* EJC may also occlude important features of the 3' SS, such as the branchpoint, polypyrimidine tract or 3' intron junction that base-pairs with the U2 snRNP complex. In the specific case of *CG7408*, we examined published EJC-CLIP [29] to reveal that the spurious 3' SS directly overlaps the location of EJC recruitment in wildtype cells (**Fig 2A**). This provides a plausible basis for testing our hypothesis. Importantly, it must be noted that this model requires removal of intron 2 prior to intron 1 (out-of-order splicing) to allow EJC masking of the cryptic 3' SS (**Fig 2G**). We tested this hypothesis by separating the cryptic 3' SS on our genomic reporter from the site of EJC recruitment, by inserting a 36 nt spacer consisting of the Flag epitope tag sequence (**Fig 2D**, genomic+Spacer). Unlike the genomic construct, which yields only annotated splice isoforms, the genomic+Flag variant yielded truncated transcripts, consistent with derepression of the cryptic 3' SS (**Fig 2F, genomic vs +Flag**). This result supports a model in which *Drosophila* EJC masks cryptic 3' SS.

However, an alternate possibility is that the Flag spacer sequence might unknowingly bear a splicing enhancer that activates the spurious 3' SS. To test this scenario, we substituted additional spacers from other epitope tags, Myc and HA. As these variants have different primary sequences, these test whether utilization of cryptic 3' SS was likely to result from unmasking, rather than enhancement. Both Myc and HA spacer variant constructs generated the same aberrantly spliced products as the Flag spacer (**Fig 2F**). Thus, our data supports a model in which the *Drosophila* EJC aids accurate SS selection during pre-mRNA processing by masking cryptic 3' SS.

We emphasize that these data support a mechanism in which intron 2 is excised first, and that out-of-order splicing mediates correct definition of the annotated intron 1 3' SS (**Fig 2G**). To gain direct transcript evidence for this, we examined recently published nanopore analysis of co-transcriptional processing (nano-COP) RNA-seq data, which was used to document that splicing order does not necessarily following the order of transcription [30]. *CG7408* lacked sufficient read depth and did not yield informative reads in nano-COP data. However, we found two genes with validated EJC-suppressed cryptic splicing (*unkempt* and *CkIIβ)* with out-of-order spliced long reads, for which out-of-order intermediately spliced products are inferred to recruit EJC to spurious 3' SS (**S4 Fig**). While out-of-order splicing has been documented as a phenomenon [30–34], it has generally been unclear if out-of-order splicing has impact on accurate pre-mRNA maturation. These experiments and analyses, along with recent work by Gehring and colleagues [10], demonstrate loci for which out-of-order splicing is critical for proper mRNA maturation.

## The EJC prevents cryptic exonic 5' SS activation during pre-mRNA processing

We next used analogous strategies to study cryptic exonic 5' SS. These sites represent ~35% of novel splice junctions upregulated under EJC-depleted conditions and are expected to be deleterious to mRNA processing fidelity. Bioinformatic analyses show cryptic 5' SS share general structural properties with 3' SS, such as clear preference in the vicinity of exon junctions but distribution across a wide range of strengths (**Figs 3A and S5A**).

We selected *CG3632* for mechanistic tests, as core-EJC knockdown activated a poorly conserved, weak cryptic 5' SS (**Figs 3B**, **S5B and S5C**–NNSPLICE score of 0.54) on exon 14. Using rt-PCR and Sanger sequencing, we validated that EJC-depletion induces a defective *CG3632* isoform lacking 71 nt of coding sequence (**Fig 3C**).

We hypothesized that the EJC, recruited to the exon 13/14 junction, suppresses the cryptic 5' SS on exon 14 and activates the canonical 5' SS during removal of intron 14. We tested this

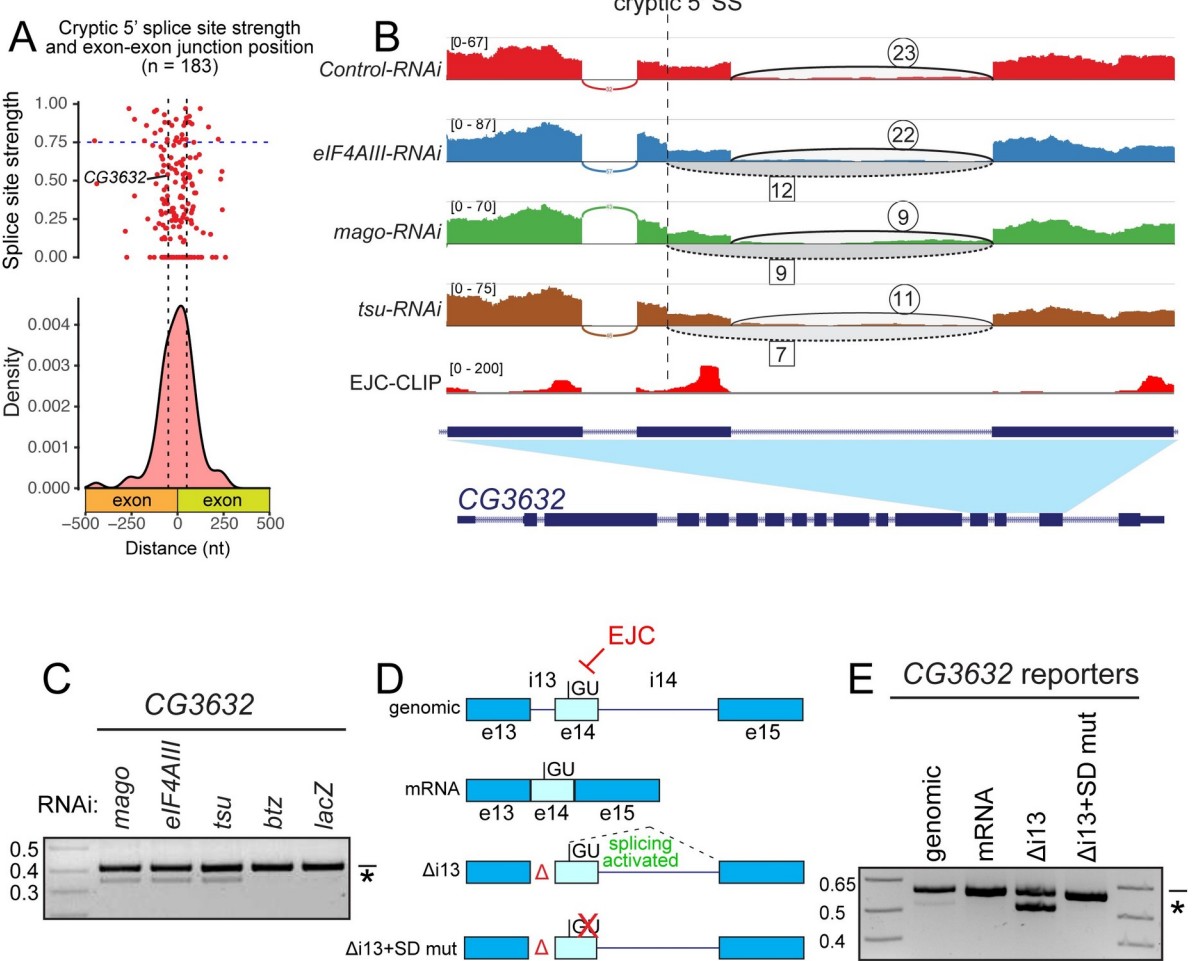

**Fig 3. EJC-depletion leads to activation of cryptic 5' splice sites.** (A) Metagene of cryptic 5' SS position relative to exon-exon boundaries as density and dot plot. The dot plot indicates splice site scores as calculated via NNSPLICE (see Materials and Methods). Horizontal dashed line depicts threshold for strong 3' SS, and vertical dashed lines specify 50 nt flanking exon-exon junctions. (B) Sashimi plot depicting HISAT2-mapped sequencing coverage along a portion of *CG3632*, which has a cryptic 5' SS that is activated under core-EJC LOF. Junction spanning read counts mapping to the canonical junction are circled, whereas cryptic junction read counts are squared. Note that spliced reads mapping to the cryptic junction are found in *eIF4AIII*, *mago* and *tsu* KD but not the control comparison. (C) Validation of *CG3632* cryptic 5' SS activation (asterisk) in core-EJC, but not *btz* or *lacZ* KD conditions. (D) Schematic of *CG3632* splicing reporters. Exons 13–15 (introns included) were cloned and subjected to further manipulation. Locations of deleted introns (Δ), as well as a construct lacking all introns (mRNA) are included. The position of the cryptic 5' SS is marked on exon 14, and was mutated in Δi13+SD mut. (E) rt-PCR of reporter (D) constructs ectopically expressed in S2 cells demonstrates that intron 13 is required for accurate processing of the minigene. Canonical products are indicated by the line and cryptic products by an asterisk.

using a minigene reporter consisting of exon 14 (containing the cryptic 5' SS) and its immediately flanking introns and exons (**Fig 3D**, genomic). Expression of this reporter in S2 cells predominantly resulted in the canonical product, but we also observed a minor amount of cryptic 5' SS activation (**Fig 3E**, genomic). As a negative control, we generated a version lacking both introns (**Fig 3D**, Δi13+14), which produced the expected mRNA (**Fig 3E**, Δi13+14). Notably, removal of intron 13 alone (**Fig 3D**, Δi13), mimicking loss of EJC recruitment at the exon 13/14 junction, yielded high levels of cryptic 5' SS activation (**Fig 3E**, Δi13) that were fully suppressed by mutation of the cryptic 5' SS in the Δi13 reporter (**Fig 3D and 3E**, Δi13+SD mut). Altogether, these data support that deposition of the EJC during pre-mRNA processing suppresses cryptic 5' SS during subsequent intron removal.

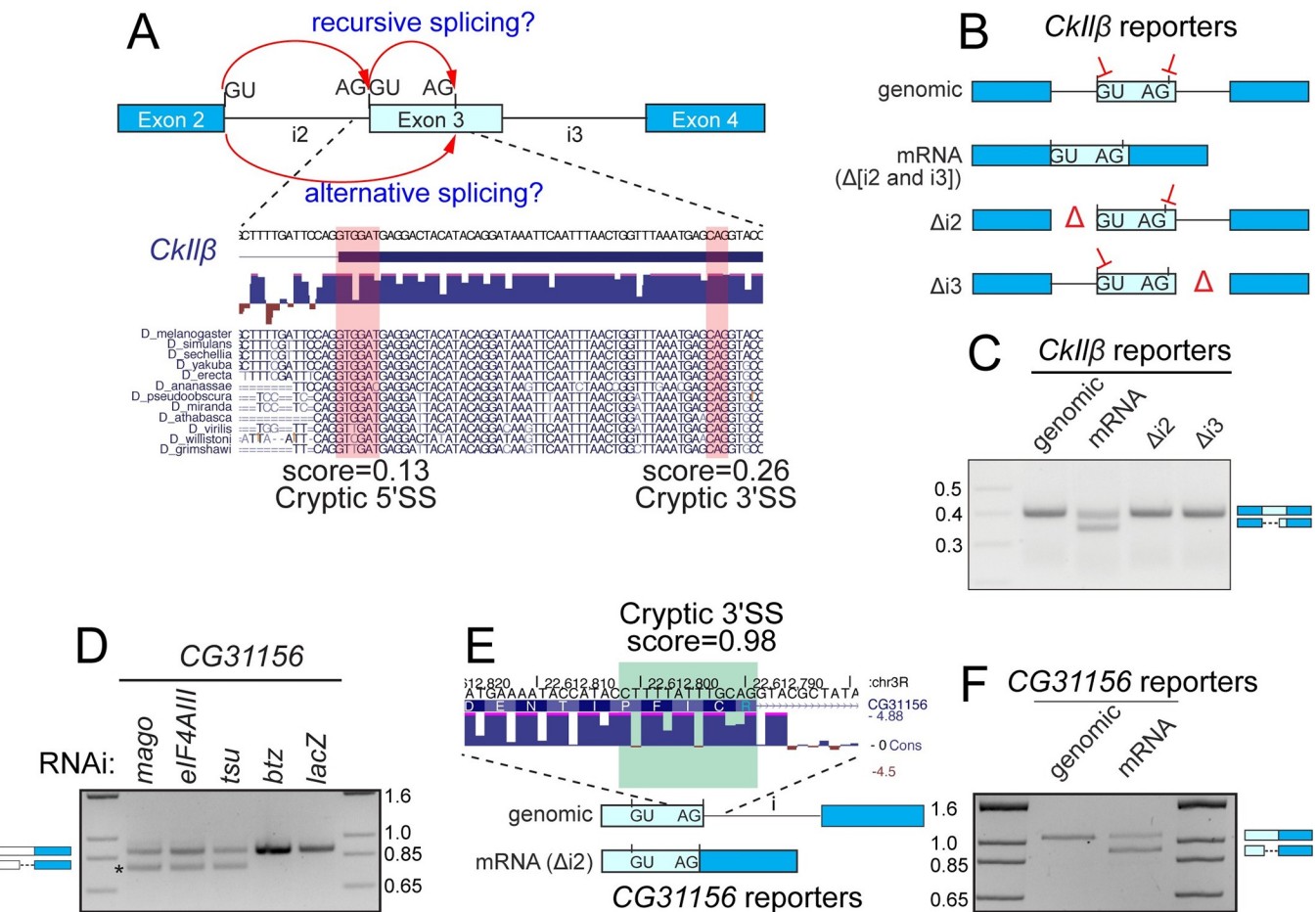

**Fig 4. EJC-depletion leads to activation of dual cryptic splice sites and resplicing of mRNA.** (A) Above: Schematic of resplicing splicing versus alternative resplicing, both of which would yield the same aberrant mRNA product. Below: Sequence of *CkIIβ* transcript lost due to cryptic splicing. Cryptic 3' SS activated is highlighted in red, as well as a potential regenerated 5' SS. Scores listed are generated by NNSPLICE. Conservation across Drosophilid family is shown. (B) Schematic of *CkIIβ* splicing reporters. Exons 2–4 (introns included) were cloned and subjected to further manipulation. Locations of deleted introns (Δ), as well as a construct lacking all introns (mRNA) are included. For reference, the position of the cryptic 3' SS and potential 5' recursive splice sites is marked on exon 3. (C) rt-PCR of *CkIIβ* reporter constructs in S2 cells demonstrates that introns are required for accurate processing of the minigene. Canonical and cryptic products are indicated. (D) Validation of *CG31156* cryptic 5' SS activation in core-EJC, but not *btz* or *lacZ* KD conditions. (E) Schematic of *CG31156* splicing reporters with and without introns. Location of potential 3' recursive splice site on exon 2 is indicated along with conservation scores. (F) rt-PCR of reporter constructs in S2 cells demonstrates that introns are required for accurate processing of the minigene. Canonical and cryptic products are indicated.

## The EJC suppresses recursive splice sites

Given that the EJC suppresses both 5' and 3' SS, a potentially more complex scenario might exist if both types of cryptic splice sites were to be activated in the vicinity of each other. We inspected our catalog of spurious junctions for this possibility, and considered that even modest matches to consensus splice sites (**Figs 2A and 3A**) might serve as viable candidates for further evaluation. Interestingly, many sequences at exon junctions were potentially able to regenerate weak splice sites after intron removal, reminiscent of the process of recursive splicing (RS) [35,36].

We first investigated a spurious junction within *Casein kinase IIβ* (*CkIIβ*), where core-EJC LOF led to loss of 54 nt of canonical mRNA sequence (**S6A and S6B Fig**). Assessment of the novel 3' SS on exon 3 revealed that it lacks a polypyrimidine tract and is a poor match to the consensus (**Fig 4A**). On the surface, the mechanism of cryptic 3' SS activation on *CkIIβ* might

appear similar to that of *CG7408* (**Figs 2F and S6D, path 2**). However, upon examining *CkIIβ* for splice sites, we found an additional poor recursive 5' SS at the beginning of exon 3 (**Fig 4A**). Therefore, we imagined an alternate scenario, whereby dual cryptic 5' and 3' SS might be derepressed upon EJC loss, leading to resplicing (**S6D Fig, path 1**). Crucially, whether one-step splicing (via alternative splicing) or resplicing (via recursive splicing event), the resulting mRNA products are indistinguishable (**Fig 4A**). Therefore, we devised reporter tests to clarify the underlying mechanism.

We first used rt-PCR to validate that core-EJC knockdown resulted in substantial activation of a truncated *CkIIβ* splice isoform corresponding to RNA-seq data (**S6C Fig**). We then analyzed a series of splicing minigenes (**Fig 4B**). Expression of *CkIIβ* exons 2–4 with all introns present produced a single product with the expected introns spliced out (**Fig 4C**, genomic). We precisely tested the positional necessity of the EJC at each exon junction by deleting each intron (**Fig 4B**, Δi2 and Δi3). These reporters also underwent normal splicing (**Fig 4C**, Δi2 and Δi3), demonstrating that *CkIIβ* processing defects were in fact mechanistically distinct from those determined for *CG7408*. Strikingly, upon testing a construct with both introns deleted, we observed a switch to truncated product output, corresponding to activation of the unannotated recursive 5' SS and 3' SS (**Fig 4B and 4C**, mRNA). This supports a model where the EJC is required at multiple positions to repress spurious 5' and 3' SS simultaneously (**S6D Fig, path 1**).

We characterized another instance of dual cryptic splice site within *CG31156*, albeit of a different flavor. Here, sashimi plots indicate activation of an exonic 5' SS within exon 2 (**S7A and S7B Fig**) and we validated this 110 nt deletion isoform using rt-PCR (**Fig 4D**). Importantly, based on these data alone, it would be reasonable to predict this as a case of alternative cryptic 5' SS activation. However, we noticed that removal of the canonical intron 2 regenerates a putative recursive 3' SS at the exon 2/3 boundary (**Figs 4E and S7C**). Therefore, we examined reporters to examine the mechanism underlying this unwanted splicing pattern. Expression of the genomic reporter that required intron removal yielded the expected mRNA product (**Fig 4F**, genomic). Conversely, deletion of the intron and expression of the mRNA resulted in the truncated re-spliced product (**Fig 4F**, mRNA). Accordingly, these data again indicate that the EJC represses dual cryptic splice sites during mRNA processing (**S7D Fig**).

## Cryptic recursive splice sites suppressed by the EJC exhibit atypical properties

We emphasize that these instances of recursive splice sites (RSS) are quite distinct from those studied previously in *Drosophila*. Fly genomes are known to contain hundreds of RSS for which the hybrid 5'/3' splice sites are highly conserved, flanked by short cryptic downstream exons, and highly biased to reside in long introns (mean length ~50 kb) [36,37]. It has been suggested that recursive splicing aids processing of long introns; however, it is also conceivable that it is easier to capture RS intermediates within long introns. The examples of cryptic RSS on the *CkIIβ* and *CG31156* transcripts clearly deviate from canonical RSS architectural properties, i.e., they are hosted in short introns and exhibit modest to poor conservation. Moreover, the example of a recursive 3' SS in *CG31156* is to our knowledge the first validated instance, and represents a conceptually novel RSS location. Importantly, the relevant AG dinucleotide in the *CG31156* 3' recursive splice site is not preserved beyond the closest species in the melanogaster subgroup (**S7C Fig**), and the amino acids encoded by the functional 5' RSS in *CkIIβ* diverge with clear wobble patterns (**Fig 4A**). Thus, these examples of cryptic exonic recursive splicing are functional, but evolutionarily fortuitous.

**The EJC protects spliced mRNAs from resplicing.**   Since many genes span large genomic regions, cDNA constructs have been a mainstay of directed expression strategies. It is generally

expected that these should be effective at inducing gain-of-function conditions, yet cDNA constructs are not typically vetted for proper processing. Our finding of dual cryptic splice sites on transcripts was alarming because in both cases, we observed resplicing on mRNA constructs (**Fig 4C and 4F**). To reiterate, the EJC prevents dual cryptic SS from resplicing on transcript segments that have already undergone intron removal, but such protection will be missing from intronless cDNA copies.

We were keen to assess the breadth of this concept. To do so, we examined the sequence of mRNAs bearing EJC-suppressed cryptic SS, and looked for additional unidentified, complementary SS. Notably, since resplicing would have to map to a canonical junction, we looked for regenerated SS at exon junction sequences. An initial survey for SS invariant dinucleotide signatures (AG for 3' SS and GT for 5' SS) indicated that 64/118 junctions with cryptic 3' SS and 104/183 junctions with cryptic 5' SS were compatible with resplicing. The fact that over half of both classes of cryptic splicing events were potentially compatible with resplicing might at first glance seem like a tremendous enrichment. However, it does in fact reflect fundamental features of extended consensus splice sequences that basepair with the spliceosome, namely the U1 snRNP and U2AF35 binding sites, respectively (**Fig 5A**-top). Quantification of these sequences indicated a range of regenerated 5' and 3' SS at exon junctions, with at least 59 junctions resembling strong SS (**Fig 5B**, NNSPLICE>0.75). However, as several cryptic 5' and 3' SS amongst our validated loci (**Figs 1–4**) were extremely poor, with functional dual cryptic splice sites in *CkIIβ* scoring at only 0.13 and 0.26 (**Fig 4A**), the functional breadth of this phenomenon is undoubtedly broader. Therefore, we imagined a scenario where a core function of the EJC is to repress splice sites that were regenerated at exon junctions as a consequence of intron removal using canonical splice sites (**Fig 5A**-bottom).

Nevertheless, as this model cannot be explicitly distinguished from alternative splicing without experimental tests, we selected additional loci for analysis. Therefore, we constructed partial cDNA constructs for three genes, encompassing regions we had validated as subject to EJC-suppression of cryptic splicing (**Fig 1C**), and selected targets that survey a range of regenerated SS strengths. These include *straw*, which yields a strong 3' RSS (NNSPLICE score of 0.98) after removal of intron 3; *eIF4G1*, which regenerates a moderate 5' RSS (NNSPLICE score of 0.64) after processing of intron 10; and *baboon*, which produces an exceptionally poor 3' RSS (NNSPLICE score of 0) after removal of intron 4, bearing only the AG dinucleotide.

In contrast to the endogenous genes which produced a single amplicon, expression of all three cDNA constructs yielded substantial re-spliced products, supporting our view that the EJC prevents activation of dual cryptic SS on mRNAs, including cryptic SS at exon junction sequences (**Fig 5C**). Unexpectedly, SS strength did not correlate with levels of re-splicing. Indeed, the majority of transcripts from all three reporters were truncated, including from *baboon*. Furthermore, the *eIF4G1* reporter yielded three truncated products, suggesting that other sequences may also serve as cryptic SS. As these examples of re-splicing occur on coding regions of the transcript, all of them either delete amino acids or generate frameshifts (**S8 Fig**).

We conclude that many cDNA constructs are potentially prone to resplicing due to loss of protection afforded by the EJC. Moreover, we propose there are distinct classes of cryptic SS within exon junction sequences. The first, examples of which were documented previously, and extended in this study, comprise strong, autonomous splice donors that occur at the 5' ends of exons and are involved in recursive splicing [10,11,35–38]. The second class, which we discover in this study, includes the auxiliary, exonic remnants of canonical splice sites subsequent to intron removal. Crucially, these are weak and are not expected to function as autonomous splice sites, but they can nevertheless become substantially activated under EJC loss-of-function conditions.

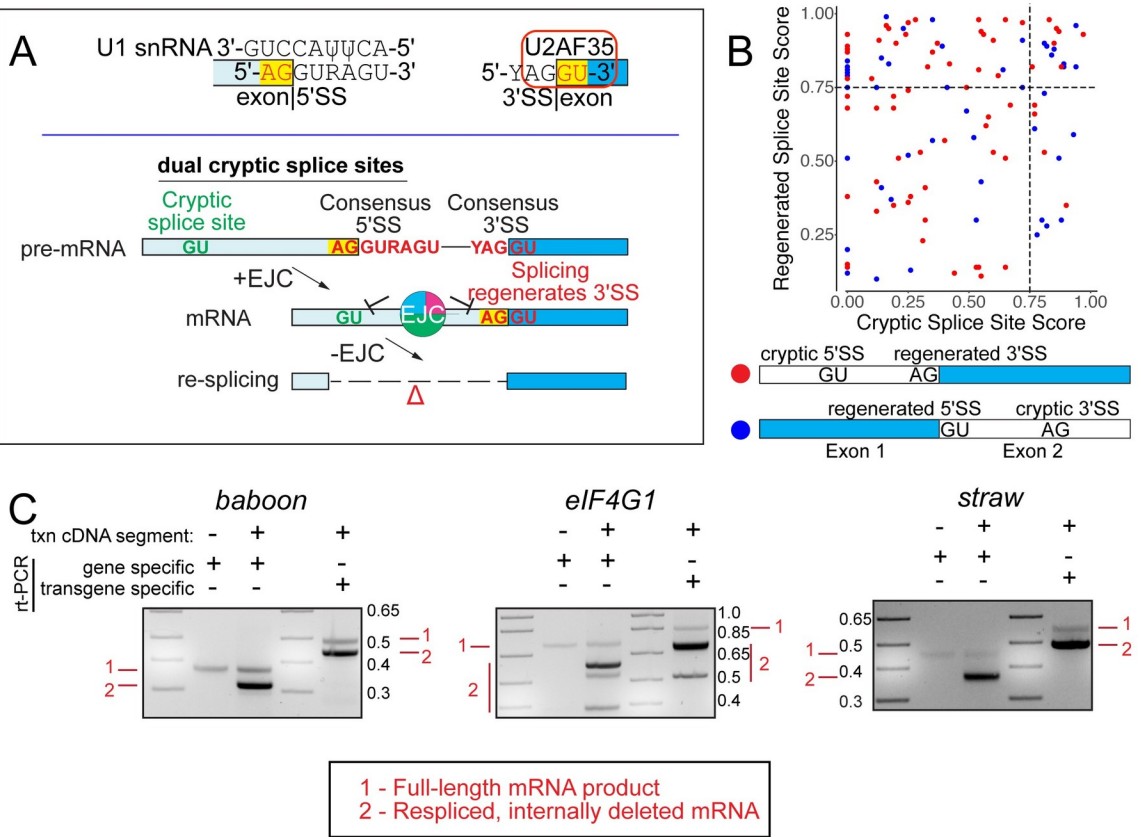

**Fig 5. Inherent features of splice sites predispose mRNA resplicing.** (A) Model for mRNA re-splicing. Top, Binding sites of U1 snRNA and U2AF35 define the 5' SS and 3' SS, respectively, but also impose constraints on flanking exonic sequences that intrinsically regenerate splice site mimics in a recursive fashion. (Bottom) When located in proximity to another cryptic splice site, these can lead to mRNA resplicing in the absence of the EJC. An example of dual cryptic splice sites with a regenerated 3' SS is shown, but this can also occur with a regenerated 5' SS. (B) Comparison of splice site strengths for cases of dual cryptic splice site activation. Cases that contain regenerated 3' and 5' splice sites at exon junctions and their structures are schematized and distinguished by red and blue dot. Dashed lines mark thresholds for reasonably strong splice sites. (C) Re-splicing on cDNAs. Constructs bearing cDNA segments of *baboon*, *eIF4G1* and *straw* were expressed in S2 cells and yielded re-spliced amplicons. Gene specific primers that can amplify both endogenous and ectopic products only show re-splicing from cells expressing cDNA reporters. To verify this directly reflects expression from the cDNA reporters, we tested amplicons that include a vector-specific primer, which yields larger bands relative to the gene-specific primers. These also demonstrate mostly re-spliced products.

## Discussion

### Conserved role for the EJC to repress cryptic splicing and its regulatory implications

Although introns are not essential for gene expression, they play important facilitatory roles by enhancing export and translation in part through recruitment of the EJC during splicing. Subsequently, it was recognized that once deposited, the EJC also promotes accurate gene expression by regulating processing of neighboring introns. Recently, in the mammalian setting, the role of the EJC during pre-mRNA splicing was extended to include suppression of cryptic splice sites [10,11].

Here we reveal that the fly EJC similarly plays a broad role in direct suppression of cryptic splice sites at exon-exon junctions, owing to its characteristic deposition. Thus, we now appreciate that concealment and suppression of cryptic splice sites is a conserved EJC activity [10]. Importantly, the positional recruitment of the EJC during splicing is conserved and sequence-

independent [18]. Thus, we infer this function should also be independent of splice site divergence between phyla, as well as splice site strength, and should not require accessory components. In contrast, non-conserved roles of the EJC appear to rely on integration within and diversification of distinct functional networks. For example, while the Upf (*Up-f*rameshift) proteins coordinate NMD across eukaryotes [39], the mechanisms differ. In mammals, NMD is coordinated with intron removal through direct interactions between the EJC and Upf3 [16,17,40]. However, these interactions are not found in invertebrates, and consequently the invertebrate EJC is not involved in NMD [19].

In addition to pre-mRNAs, we show that the EJC also suppresses cryptic splice sites within spliced mRNAs. Although this mechanism cannot be distinguished from alternative splicing (**Fig 4A**) without directed experimentation, we readily detect re-splicing on all cDNA constructs tested. Unexpectedly, while these junctions appear to contain just one cryptic SS, our data indicates that these transcripts contain secondary cryptic splice sites that mediate resplicing. Importantly, we validate that even poor matches to SS consensus motifs are competent for re-splicing. Curiously, as all of our demonstrated examples involve a recursive event at either the 5' or 3' cryptic SS, our findings broaden a phenomenon that was previously described within long introns [36,37]. Furthermore, canonical SS sequences that undergo base pairing interactions with U1 snRNA (5' SS) and U2AF35 (3' SS) have motifs AG|GURAGU and YAG|GU [5,41]. It is noteworthy that core splice site signals contain bases that are compatible with regeneration of splice sites and that these naturally occur proximal to EJC recruitment sites. Accordingly, we propose that an ancestral function for the conserved position of EJC deposition may be to prevent accidental activation of regenerated splice sites.

Finally, our observations of re-splicing on cDNAs reflect an essential function for introns in protecting mRNA fidelity. For all tested cases of cDNA resplicing on coding sequences, we note deletions of peptide segments or truncations with loss of domains required for protein function. Importantly, these affected targets include essential genes, such as *eIF4G1* and activin receptor *baboon*. In the case of *baboon*, the 54 nt splicing defect leads to a deletion of 18 amino acids (195–212, **S8A Fig**). For *eIF4G1*, re-splicing removes 131 nt of mRNA sequence, alters the open reading frame and leads to protein truncation with loss of the MI and W2 domains (**S8B Fig**). Finally, re-splicing on *straw* transcripts also alters reading frame by removing 91 nt of mRNA, and is predicted to remove 2/3 Plastocyanin-like domains (**S8C Fig**). Thus, our findings have serious implications for functional genomics as well as community genetic studies [42,43], where cDNA expression constructs and collections are often employed with little attention paid to mRNA processing. Altogether, our work uncovers important functions for intron removal and the role of the EJC to protect the transcriptome from unwanted re-splicing.

## Materials and methods

### Bioinformatic analysis

**Datasets.** We obtained several published datasets from the NCBI Gene Expression Omnibus or the European Nucleotide Archive for analyses in this study. Core-EJC knockdown RNA-sequencing datasets were reported by the Roignant lab [28]: GSE92389. EJC CLIP datasets were published work by the Ephrussi lab [29]: PRJEB26421. Raw sequencing data was mapped to the *Drosophila* reference genome sequence (BDGP Release 6/dm6) using HISAT2 [44] under the default settings. Direct chromatin RNA nanopore data (nano-COP) from S2 cells was reported by the Churchman lab [30]: GSE123191. We mapped nano-COP data to the *Drosophila* reference genome sequence (BDGP Release 6/dm6) using minimap2 with parameters -ax splice -uf -k14.

**Splicing analyses.** Splice junctions were mapped using the MAJIQ algorithm (2.0) under default conditions [45]. Splice graphs and known/novel local splice variants were defined with the MAJIQ Builder using annotations of known genes and splice junctions from Ensembl release 95 and all BAM files. The MAJIQ Quantifier was used to calculate relative abundances (percent selected index—PSI) for all defined junctions. The resulting data was output into tabular format using the Voila function.

A custom R script was written to process all MAJIQ-defined novel junctions relative to the Ensembl gene annotations and identify *de novo* EJC-suppressed junctions. First, we quantified usage of all novel junctions by mining mapped libraries (BAM files) for high quality junction spanning reads with at least 8 nt of overhang and no mismatches. These counts were normalized to sequencing depth per library. To identify *de novo* junctions that may be upregulated, we first selected junctions with at least 5 split reads. In order to enrich for *de novo* junctions that are suppressed by the EJC pathway, we looked for those with > 2 fold difference in at least 2/3 core-EJC RNAi conditions relative to the *lacZ* control. To apply further stringency, we also required that the PSI measurements reflect sufficient change between treatment and control conditions. Therefore, we applied an additional filter of PSI fold change > 2 in at least 2/3 core-EJC RNAi conditions. These criteria produced a total of 573 novel junctions. All junctions are reported in **S1 Table**.

The 5' and 3' ends of these junctions were compared against known gene annotations to characterize splice sites. Exonic 5' and 3' SS reflect sites that mapped on exons while the other end mapped to a canonical splice junction, and the same process was used to define intronic 5' and 3' SS. *de novo* cases of alternative splicing reflect junctions that utilized annotated splice sites but represented novel connectivity. Sashimi plots were generated using features available on the Integrative Genomics Viewer (IGV) [46].

We generated a custom pipeline to assess recursive splicing potential (Fig 4). Briefly, we identified transcripts that contained cryptic exonic 5' and 3' splice sites. For these transcripts, we mapped the position of all splice junctions on the mRNA, which could in theory generate the observed splicing defects. We examined sequences directly downstream of relevant splice junctions to identify potential 5' recursive splicing and those directly upstream to identify potential 3' recursive splicing.

We calculated splice site strengths using NNSPLICE (https://www.fruitfly.org/seq_tools/splice.html) [47]. The sequences used for these analyses were obtained from mRNA rather than the genomic context, which may contain intronic sequences as well. To generate nucleotide content plots, splice sites and their indicated flanking sequences were obtained from mRNAs and fed to WebLogo version 2.8.2 [48]. The splice sites are centered in these plots.

**Branchpoint analysis.** We sought to identify possible branchpoints upstream of spurious 3' SS. We first derived a BP position weight matrix (PWM) by calling motifs (using MEME) on sequences 15–45 nt upstream of 3' SS from 10,000 randomly selected introns from *Drosophila*. This strategy has previously been adopted to identify putative branch point motifs [49,50]. We then used the obtained PWM to find potential BP sequences upstream of spurious 3' SS using a minimum of 75% PWM match.

## Constructs and cell culture

All splicing reporters were cloned into pAC-5.1-V5-His (ThermoFisher Scientific) using compatible restriction sites. We used PCR to amplify minigene splicing reporters from Drosophila genomic DNA, and used site directed mutagenesis to remove specified introns. We used cDNAs to amplify reporters lacking introns. For genes with multiple isoforms (such as *CG7408*), we cloned the dominant fragment. All primers used for generating constructs and mutagenesis have been summarized in **S3 Table**.

Transfections were performed using S2-R+ cells cultured in Schneider *Drosophila* medium with 10% FBS. Cells were seeded in 6-well plates at a density of $1 \times 10^6$ cells/mL and transfected with 200 ng of plasmid using the Effectene transfection kit (Qiagen). Cells were harvested following 3 days of incubation.

## Knockdown of EJC factors in S2 cells

The indicated EJC components were knocked down via RNAi (dsRNA-mediated interference) in S2-R+ cells. The MEGAscript RNAi kit (ThermoFisher Scientific) was used to produce dsRNAs required for this experiment. Briefly, DNA templates containing promoter sequences on either 5' end were produced through PCR with T7-promoter-fused primers. 2 μg of DNA template was transcribed *in vitro* for 4 hours as recommended by the manufacturer. The products were incubated at 75˚ C for 5 minutes and brought to room temperature to enhance dsRNA formation. A cocktail of DNaseI and RNase removed DNA and ssRNAs, and the remaining dsRNA was purified using the provided reagents. All dsRNA reagents were verified by running on a 1% agarose gel and quantified by measuring absorbance at 260 nm using a NanoDrop (ThermoFisher Scientific).

For knockdown, $3 \times 10^6$ S2-R+ cells in 1 mL serum free medium were incubated with 15 μg of dsRNA for 1 hour at room temperature. Then, 1 mL of medium containing 20% FBS was added to the cells and the whole mixture was moved to a 6 well plate. Cells were collected after 4 days of incubation.

## RT-PCR

After transfection or RNAi treatment, cells were washed in ice cold PBS and pelleted using centrifugation. RNA was collected using the TRIzol reagent (Invitrogen) under the recommended conditions. 5 μg of RNA was treated with Turbo DNase (Ambion) for 45 min before cDNA synthesis using SuperScript III (Life Technology) with random hexamers. RT-PCR was performed using AccuPrime Pfx DNA polymerase (ThermoFisher Scientific) with standard protocol using 26 cycles and primers that were specific to each minigene construct. All primers are listed and described in **S3 Table**.

## Supporting information

**S1 Fig. Core-EJC depletion yields broad activation of *de novo* splice junctions.** (A) Strong overlap of de novo splice junctions between core-EJC knockdown conditions. The Venn diagram depicts which of 1677 junctions with at least 5 split reads had > 2-fold split read changes between treatment and controls. p-value for three-way overlap was calculated using a permutation test with 10^8 tests. (B) Strong overlap of high-confidence de novo splice junctions between core-EJC knockdown conditions. The Venn diagram depicts which of 876 junctions with at least 5 split reads and > 2-fold split read changes also show > 2-fold changes in percent selected index (PSI) between treatment and controls. *p*-value for three-way overlap was calculated using a permutation test with 10^8 tests. (C) Knockdown of EJC factors in S2 cells using dsRNA. quantitative rt-PCR of core-EJC and *btz* transcripts after dsRNA treatment. (D) Sashimi plot illustrating the expression of annotated alternative isoforms for unkempt (unk) in S2 cells. The location of rtPCR primers and alternative isoforms are included.
(TIF)

**S2 Fig. Locations of rtPCR primers within target transcripts.** Gene models for genes tested in Fig 1C and primers are illustrated from the UCSC genome browser. Mapping of products

obtained from spurious splicing indicate transcript changes.
(TIF)

**S3 Fig. A majority of cryptic 3' SS activated under EJC-loss are weak.** (A) Nucleotide content of cryptic 3' SS. These sequences, apart from the invariant AG dinucleotide show poor strength. (B) BP motif obtained by analysis of 10000 annotated introns. (C) Plot indicating the relationship between BP motif and spurious 3' SS strength. (D) Example of a weak cryptic 3' SS (NNSPLICE score of 0.29) found on the *CG7408* transcript. Sashimi plot indicating splice junction counts in EJC LOF and control datasets. The locations of annotated and spurious 3' SS are shown. Conservation of the weak splice site is depicted using the multiple alignment format on the UCSC genome browser, as well as phyloP and phastCons scores. The location of spacer sequences schematized in Fig 2D is marked.
(TIF)

**S4 Fig. Evidence for out-of-order intron removal in *unkempt* and *CkIIβ*.** (A) Evidence for out-of-order intron removal for *unkempt*. Top: Sashimi plot indicating the expression of annotated and spurious splicing using control and *mago* knockdown RNA sequencing datasets. The location of the spurious 3' SS relative to the wildtype transcript isoforms is marked. Below: nano-COP reads mapping to the portion of *unkempt* that undergoes spurious splicing. Within the reads, boxes represent coverage and lines represent skipped coverage due to splicing. Note that skipped read coverage maps to the location of annotated introns. Reads that indicate out-of-order processing are marked–here the downstream intron is removed, but an upstream intron (according to the annotated models) is not processed. Within this locus, there were no examples of reads with all introns removed. (B) Evidence for out-of-order intron removal for *CkIIβ*. As in (A), the top shows a sashimi plot whereas the bottom represents nano-COP reads. For this locus, there was evidence for fully spliced, fully unspliced and partially spliced products. We identified two instances of out-of-order intron removal and one instance of ordered intron removal.
(TIF)

**S5 Fig. A majority of cryptic 5' SS activated under EJC-loss are weak.** (A) Nucleotide content of cryptic 5' SS. (B) Schematic of a *de novo* splicing event detected on the *CG3632* transcript. Validation of splicing defects shown on the right. (C) Cryptic 5' SS (NNSPLICE score of 0.54) found on the *CG3632* transcript. Conservation of the weak splice site is depicted using the multiple alignment format on the UCSC genome browser, as well as phyloP and phastCons scores.
(TIF)

**S6 Fig. *de novo* splicing on *CkIIβ* is a result of dual cryptic splice site activation.** (A) Sashimi plot depicting HISAT2-mapped sequencing coverage along a portion of *CkIIβ*, which has a cryptic 3' SS that is activated under core-EJC LOF. Junction spanning read counts mapping to the canonical junction are circled, whereas cryptic junction read counts are squared. Note that spliced reads mapping to the cryptic junction are found in *eIF4AIII*, *mago* and *tsu* but not the control comparison. (B) Schematic of a *de novo* splicing event detected on the *CkIIβ* transcript. (C) Validation of *CkIIβ* cryptic 3' SS activation in core-EJC, but not *btz* or *lacZ* KD conditions. (D) Models that explain the *CkIIβ* splicing defects. Path 1 and 2 reflect alternate orders of intron removal. Crucially, path 1 leads to EJC-suppressed cryptic splicing on mRNAs using the indicated 5' recursive splice site and a cryptic 3' SS, whereas path 2 can also produce a splice defect after removal of intron 2.
(TIF)

**S7 Fig. *de novo* splicing on *CG31156* is a result of dual cryptic splice site activation.** (A) Sashimi plot depicting HISAT2-mapped sequencing coverage along a portion of *CG31156*, which has a cryptic 5' SS that is activated under core-EJC LOF. Junction spanning read counts mapping to the canonical junction are circled, whereas cryptic junction read counts are squared. Note that spliced reads mapping to the cryptic junction are found in *eIF4AIII*, *mago* and *tsu* but not the control comparison. (B) Schematic of a *de novo* splicing event detected on the *CG31156* transcript. (C) Conservation of the cryptic 5' SS (NNSPLICE score of 0.54) and a potential 3' recursive splice site (NNSPLICE score of 0.98) found on the *CG31156* transcript highlighted in green, relative to the gene model. Conservation of the splice site is depicted using the multiple alignment format on the UCSC genome browser, as well as phyloP scores. Canonical splice sites are highlighted in yellow. (D) Model of activation of dual cryptic splice sites on the *CG31156* transcript. Activation of the cryptic 5' SS with an additional cryptic 3' recursive splice sites leads to deletion of 110 nt of mRNA.
(TIF)

**S8 Fig. Re-splicing on mRNAs alters translated proteins.** (A-C) Protein and transcript structures are schematized and the location of cryptic resplicing highlighted in blue. (A) Re-splicing on *baboon* leads to a 54 nt deletion of the mRNA and an 18 amino acid deletion. The deletion does not overlap known domains. Conservation plots for deleted 54 nt region is included. (B) Re-splicing on *eIF4G1* leads to a 131 nt deletion, leading to a change in reading frame and truncation of the C terminal domains of *eIF4G1*. Importantly, critical domains required for *eIF4G1* function are lost due to re-splicing. (C) Re-splicing on *straw* leads to a 91 nt deletion, leading to a change in reading frame and truncation of the protein. Importantly, 2 of 3 Plastocyanin-like domains are lost due to transcript defects.
(TIF)

**S1 Table. *Drosophila* cryptic splice junctions that are repressed by the Exon Junction Complex (EJC).** This table provides the genomic locations, host genes, and classification of de novo unannotated splice junctions discovered in EJC-knockdown data from S2 cells.
(XLSX)

**S2 Table. Gene ontology (GO) analysis of genes with cryptic splicing that is repressed by the EJC.**
(XLSX)

**S3 Table. Oligonucleotide sequences used in this study.** The tabs summarize primers used to clone constructs, to knockdown gene products, and to assess mRNA processing.
(XLSX)

## Author Contributions

**Conceptualization:** Brian Joseph, Eric C. Lai.

**Data curation:** Brian Joseph.

**Formal analysis:** Brian Joseph.

**Funding acquisition:** Eric C. Lai.

**Methodology:** Brian Joseph.

**Project administration:** Eric C. Lai.

**Resources:** Brian Joseph.

**Writing – original draft:** Brian Joseph, Eric C. Lai.

**Writing – review & editing:** Brian Joseph, Eric C. Lai.

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
