## [Decision Letter · Decision Letter 0]

4 Dec 2020

Dear Dr Lai,

Thank you very much for submitting your Research Article entitled 'The Exon Junction Complex and intron removal prevents re-splicing of mRNA' to PLOS Genetics.

The manuscript was fully evaluated at the editorial level and by independent peer reviewers. While both reviewers found value in examining the conservation of splicing regulation by the exon-junction complex in drosophila, there were several experimental and computational issues that need to be addressed. Moreover, there were substantial concerns about the interpretation and the presentation of the data, which does not sufficiently frame how these new findings fit into the context of what is already known about the connections of the EJC to splicing or what is novel in your work compared to prior studies. Based on the reviews, we will not be able to accept this version of the manuscript, but we would be willing to review a much-revised version. We cannot, of course, promise publication at that time.

If you decide to revise the manuscript for further consideration at PLOS Genetics, please aim to resubmit within the next 60 days, unless it will take extra time to address the concerns of the reviewers, in which case we would appreciate an expected resubmission date by email to plosgenetics@plos.org.

[LINK]

We are sorry that we cannot be more positive about your manuscript at this stage. Please do not hesitate to contact us if you have any concerns or questions.

Yours sincerely,

Teresa Bowman

Guest Editor

PLOS Genetics

John Greally

Section Editor: Epigenetics

PLOS Genetics

Reviewer's Responses to Questions

**Comments to the Authors:**

Reviewer #1: The authors have reanalyzed RNA-seq data, previously published by Akhtar et al. 2019, for EJC knockdowns in Drosophila cells. They follow with RT-PCR of specific transcripts and from cells transfected with reporter constructs to provide experimental validation. Overall, they conclude that loss of EJC deposition results in activation of cryptic splice sites, as was found in human cells (Gehring, Mol. Cell 2018). The issue of the extent to which the EJC influences splicing is an interesting one, and I value such data from more than one model system. However, I am not convinced that the data in the current manuscript are sufficiently compelling on their own for many of the conclusions drawn. Rather, they are consistent with an interpretation already made in the human system.

Overall, the writing needs to be more precise, the presentation of the data more complete, and the description of what can and cannot be concluded from the data more accurate.

Specific Comments:

1 -p.2. I do not understand how the authors can justify saying “Unexpectedly, we discover the EJC inhibits scores of regenerated 5' and 3' recursive splice sites on segments that have already undergone splicing”, when this is exactly one of the findings of the Gehring Mol Cell paper.

2 -pp.3-5. The Introduction is overall unclear, muddled, and confusing. The last paragraph is clear, but otherwise the Introduction does not sufficiently represent the history of the EJC in splicing.

3 -p.3. “Cryo-EM structures of prespliceosomal complexes show that U1 snRNA establishes base contacts across the -2 to +6 position for a typical 5’SS…”. This is a rather bizarre statement. It is true, but it seems to assert that U1-5’SS base pairing was discovered by cryo-EM, rather than proven by biochemistry and genetics over 30 years ago.

4 -Fig. 1A. This schematic doesn’t make much sense at first. It is not clear that the blue/black represents the canonical WT splicing isoform, and the labeled splice sites and red lines represent only cryptic sites and splicing.

5 -Fig. 1C, and 2C,E,F. What exactly are these alternative isoforms? There should be Sanger sequencing information of these PCR products provided.

6 -Likewise, what are any of these amplicons supposed to be? The primer binding sites should be shown in a schematic for each gene. Without this information, it is quite difficult to evaluate any of these data.

7 -1C panel 7 is missing an asterisk for the cryptic product.

8 -All the gels have DNA ladders included, but none are labeled, so the reader doesn’t know what size any of the bands are.

9 -p.7. “spurious exonic 3' SS “ and “cluster specifically around exon junctions”, etc. The authors need to adopt more precise language. For the RNA molecule in which the cryptic 3’SS is used, it is the 3’SS and it is not exonic. The authors are comparing to a canonical splicing pattern found in WT cells; they need to be more precise to avoid confusion.

10 -p.8. “Our cryptic junction replaces intron 1”. How can a junction replace an intron?

11 -p.8. “this reporter recapitulated normal splicing through activation of annotated 3' SS”. To what control is this new reporter compared to show that it recapitulates “normal” splicing? No such control is shown.

12 -p.8. “At face value, this appears consistent with the hypothesis that the EJC regulates splicing of flanking introns.” Huh?? This conclusion does not follow from the previous sentences. Perhaps the authors thought they put it elsewhere?

13 -p.8. Language like “pre-processed exon junctions” and “pre-spliced” is confusing and inaccurate.

14 -p.8 and Figure 2 D, E: There is no evidence here that this is related to EJC deposition. An alternative model, not considered, is that i2 deletion brings an exonic enhancer present in e3 close to the cryptic 3’SS found in e2.

15 -Likewise the authors have no evidence for the order of intron removal in CG7408.

16 -Fig. 2F. What is the 36-nt sequence that was added? This is not provided. How do we know that it is a ‘neutral’ sequence? Could it contain a splicing enhancer? (Honestly, it’s not so easy to come up with a neutral sequence.)

17 -p.9 and Fig 2A and 3A. “clear preference in the vicinity of exon junctions but distribution across a wide range of strengths.” What is the X-axis? The vast majority of exons are not 500 nt long. Most are ~100-150 nts, so this schematic just shows that the cryptic 5’SS are found within a distance of a typical exon from the WT 5’SS. It looks like the authors used -500 and +500 sequence from the selected WT exon junction, but these are not necessarily all exonic sequence; however, it is not at all clear what sequence the authors use here.

18 -There are logical inconsistencies. For example, on p.15 the authors correctly admit the they cannot distinguish recursive splicing from alternative splicing; yet, in the next sentence state that they “readily detect re-splicing on all cDNA constructs tested”.

19 -p.16. “…our work uncovers an important co-transcriptional function of intron removal…”. I do not see any data that address the co-transcriptional nature of the current findings.

Typos

20 -p.3. “nucletodies” should be “nucleotides”.

21 -p.15. “experimentaion” should be “experimentation”.

Reviewer #2: The authors investigate whether the recently reported function of the EJC in suppressing cryptic SS is conserved in fly. They do this by initally performing a reanalysis of previously published EJC component KDs + RNA-seq, finding substantial activation of cryptic SS shared across KDs of the three EJC core proteins. They use RT-PCR to validate a number of these cryptic SS, and then proceed to explore the mechanism of these changes using minigenes for a select few genes. For both 5' and 3' SS they present convincing evidence that the EJC normally blocks recognition of the cryptic SS. They next use similiar minigene experiments to show some proportion of the cryptic events correspond to resplicing, and point out that this is potentially quite common given the canonical SS motifs (i.e. the frequent GT at the end of the exon and the AG at the beginning). They point out that this is concerning for transgene experiments where the processed sequence is introduced since this will not have the benefit of the EJC protecting it from resplicing.

The paper is well written, motivated and clear. It would be helpful to label the exon/intron numbers in all figures to make them easier to connect to the text.

No discussion is given to the strength of the branchpoint sequence for the regenerated introns. It is surprising that such weak (e.g. 0 NNSPLICE scoring) 3' SS are splice-component: is this being compensated by strong BP? If not, what is the authors' explanation for the splicing of such weak SS?

Minor: I'm confused about the statement that "spurious exonic 3' SS. These represent a majority...". From fig 1 it looks like exonic 5' SS are the biggest category?

Overall this is a nice piece of work with sensible bioinformatic analysis, convincing experiments, and important findings for both splicing biology and functional genetics more broadly.

**Have all data underlying the figures and results presented in the manuscript been provided?**

Reviewer #1: Yes

Reviewer #2: None

PLOS authors have the option to publish the peer review history of their article (what does this mean?). If published, this will include your full peer review and any attached files.

Reviewer #1: No

Reviewer #2: No

---

## [Decision Letter · Decision Letter 1]

8 Apr 2021

Dear Dr Lai,

Thank you very much for submitting your Research Article entitled 'The Exon Junction Complex and intron removal prevent re-splicing of mRNA' to PLOS Genetics.

The manuscript was fully evaluated at the editorial level and by independent peer reviewers. The reviewers appreciated the attention to an important topic but identified some concerns that we ask you address in a revised manuscript. The comments are mostly to improve the description in the text and in the figures as to the identity of the tested amplicons. This point is important to address to ensure the ability of future interested scientists to reproduce the work. 

We therefore ask you to modify the manuscript according to the review recommendations. Your revisions should address the specific points made by reviewer #1. The revised work can be assessed at the editorial level to help expedite a final decision. 

[LINK]

Yours sincerely,

Teresa Bowman

Guest Editor

PLOS Genetics

John Greally

Section Editor: Epigenetics

PLOS Genetics

Reviewer's Responses to Questions

**Comments to the Authors:**

Reviewer #1: Overall, this version is much improved. In particular, the introduction is much more clear, coherent, and complete. The results also are better described and the language more clear. However, a few aspects still need clarification.

1. Figure 1. I appreciate the improved clarity of the model and the addition of the size markers in 1C. However, as far as I can see there are still no indicators as to what the amplicons in the gels are. For example, icons depicting the amplicons like those used in Figure 4D and F would be helpful – except please include an indication of what exons are represented.

2. This also relates to Question 5, which was “what exactly are the amplicons?” The authors say that they confirmed amplicons by Sanger sequencing or by size, and I’m mostly ok with that (although they do not tell us which amplicons were confirmed by sequencing), but they don’t tell us what the amplicons actually are. Without information as to the identity of the amplicons, this work could never be replicated.

3. In Figure 4F, the amplicon icons on the right seem too high, i.e. they aren’t aligned with the gel bands.

4. Question 19. The authors say that they re-worded their statement about “...our work uncovers an important co-transcriptional function of intron removal...”. I maintain that their work does not address the co-transcriptionality of the current findings – and the authors say that they agree. However, on page 19 in the last sentence of the Discussion, they still say “our work uncovers an important co-transcriptional function of intron removal”.

5. Figure 5 C. Again, it would really help the reader to have explicit schematic or icon next to the gels that show what the amplicons are. It is also unclear why the bands using the transgenic vs genetic specific are so different in size.

6. On p. 34, Figure 5 Legend is labeled incorrectly. There are 2 legends labeled "Figure 4".

Reviewer #2: I was already positive about this manuscript and the authors have improved it on revision. I appreciate them also having taken the time to look into whether BP sequence might account for some of the variation they see. The other reviewer was more critical so I will leave it to them to determine whether their concerns are addressed - it certainly appears the authors have made substantial efforts to do so.

**Have all data underlying the figures and results presented in the manuscript been provided?**

Reviewer #1: None

Reviewer #2: None

PLOS authors have the option to publish the peer review history of their article (what does this mean?). If published, this will include your full peer review and any attached files.

Reviewer #1: No

Reviewer #2: No

---

## [Editor Report · Decision Letter 2]

26 Apr 2021

Dear Dr Lai,

We are pleased to inform you that your manuscript entitled "The Exon Junction Complex and intron removal prevent re-splicing of mRNA" has been editorially accepted for publication in PLOS Genetics. Congratulations! Thank you for carefully addressing all reviewers comments. 

Yours sincerely,

Teresa Bowman

Guest Editor

PLOS Genetics

John Greally

Section Editor: Epigenetics

PLOS Genetics

Comments from the reviewers (if applicable):

**Data Deposition**

http://datadryad.org/submit?journalID=pgenetics&manu=PGENETICS-D-20-01536R2

**Press Queries**

---

## [Editor Report · Acceptance letter]

21 May 2021

PGENETICS-D-20-01536R2 

The Exon Junction Complex and intron removal prevent re-splicing of mRNA 

Dear Dr Lai, 

We are pleased to inform you that your manuscript entitled "The Exon Junction Complex and intron removal prevent re-splicing of mRNA" has been formally accepted for publication in PLOS Genetics! Your manuscript is now with our production department and you will be notified of the publication date in due course.

With kind regards,

Katalin Szabo

PLOS Genetics

On behalf of:
